# Magnetic Sulfonated Melamine-Formaldehyde Resin as an Efficient Catalyst for the Synthesis of Antioxidant and Antimicrobial Pyrazolone Derivatives

**Shefa Mirani Nezhad** [1], **Seied Ali Pourmousavi** [1], **Ehsan Nazarzadeh Zare** [1,*], **Golnaz Heidari** [2] and **Pooyan Makvandi** [3]

[1] School of Chemistry, Damghan University, Damghan 36716-41167, Iran; shefamirani@yahoo.com (S.M.N.); pourmousavi@gmail.com (S.A.P.)

[2] Department of Chemistry, Faculty of Science, University of Guilan, Rasht 41938-33697, Iran; golnazheidari68@yahoo.com

[3] Istituto Italiano di Tecnologia, Centre for Materials Interfaces, Viale Rinaldo Piaggio 34, 56025 Pontedera, Italy; pooyanmakvandi@gmail.com

\* Correspondence: ehsan.nazarzadehzare@gmail.com or e.nazarzadeh@du.ac.ir

**Abstract:** Sulfonated polymer-based materials, among heterogeneous catalysts, are frequently utilized in chemical transformations due to their outstanding chemical and physical durability. In this regard, a magnetic sulfonated melamine–formaldehyde resin (MSMF) catalyst was successfully prepared from a mixture of sulfonated melamine–formaldehyde and $Fe_3O_4$ nanoparticles in two steps. MSMF was used as a heterogeneous catalyst for the one-pot, three-component condensation of benzyl pyrazolyl naphthoquinones in water as a green solvent and 4-[(indol-3-yl)-arylmethyl]-1-phenyl-3-methyl-5-pyrazolones. The antimicrobial and antioxidant activities of catalyst, benzyl pyrazolyl naphthoquinones, and 4-[(indol-3-yl)-arylmethyl]-1-phenyl-3-methyl-5-pyrazolones were evaluated using agar disk-diffusion and DPPH assays, respectively. The antioxidant activity of the catalyst and 4-[(indol-3-yl)-arylmethyl]-1-phenyl-3-methyl-5-pyrazolones was found to be 75% and 90%, respectively. Furthermore, catalyst, benzyl pyrazolyl naphthoquinones, and 4-[(indol-3-yl)-arylmethyl]-1-phenyl-3-methyl-5-pyrazolones exhibited antimicrobial activity against *Staphylococcus aureus* and *Escherichia coli*. In conclusion, MSMF is a superior catalyst for green chemical processes, owing to its high catalytic activity, stability, and reusability.

**Keywords:** multicomponent reactions; heterogeneous catalysis; magnetic sulfonated melamine–formaldehyde; pyrazole derivatives; polymer-based catalysts

## 1. Introduction

Mineral and organic acids have been widely used in various chemical and industrial processes, such as fertilizer manufacturing, mineral processing, and detergent production, as well as electrolytes, catalysts, and industrial cleaning agents. The most significant disadvantage of using liquid acids as catalysts or cocatalysts in chemical processes is that they make end-product recovery more difficult by requiring additional purification, separation, and detoxification procedures to remove unreacted acids [1]. As a result of these constraints, research has accelerated on recyclable strong solid acids, such as $H_2SO_4$, as an environmentally benign and green alternative to non-recyclable liquid acids. One method is to use mesoporous materials, polymers, nanomagnetites, and metal oxides as heterogeneous acid catalysts. To synthesize an effective heterogeneous acidic catalyst, a combination of qualities is required, such as high acid strength, dispersed acid sites, high acid concentration, and an extended nanoporous structure with a large surface area. For example, nanomaterial-based acid catalysts exhibit greater selectivity and activity, despite

reactants having limited access to the inner active sites of acid catalysts. This is due to a significant increase in the surface area of the supported system, which improves the interaction of reactant molecules with the catalyst [2].

Solid acid catalysts have received much attention, compared with homogeneous acids, due to their easy separation and reusability. Solid acid catalysts, compared with their homogeneous acid counterparts, are less active and more expensive. However, they have advantages, such as their handling requirements, simplicity, the versatility of process engineering, catalyst regeneration, decreased reactor and plant corrosion problems, and environmentally safe disposal [3]. Most efforts have been directed toward the preparation of a solid, thermally stable catalyst in which the acid sites have the required acidity (type and strength) to catalyze a given type of chemical reaction [4]. Recent studies have proven the technical feasibility, as well as the environmental and economic benefits, of biodiesel production via heterogeneous acid-catalyzed esterification and transesterification [5]; for example, sulfonated graphitic carbon nitride was used to synthesize biodiesel [6]. Sulfonated dendritic mesoporous silica nanospheres were used in a metal-free Lewis acid catalyst for the upgrading of carbohydrates [7].

With several uses as efficient catalysts in organic processes in diverse fields, the application of solid acid polymer-based catalysts has expanded in recent years [8]. Among them, melamine–formaldehyde resin (MFR), one of the most effective thermoset polymers, has outstanding physical and chemical stability due to the use of irreversible covalent bonds to interlink building units rather than weak coordination or reversible covalent links. More significantly, compared with other polymer-based solid acid catalysts made from costly monomers, it can be easily manufactured from inexpensive and readily available monomers. Furthermore, MFR has a considerable number of secondary amine nitrogen atoms that can be employed to immobilize acidic functions. Methylolation and condensation are the two processes used in the production of this amino resin. Heat and boil resistance, transparency, corrosion resistance, improved toughness, impermeability, surface smoothness, and fire safety are some of the benefits of MFR when used in major industrial applications. MFR was first applied in wood glue but is currently utilized in coatings, ornamental and floor laminates, adhesives, and molding compositions [9]. Compared with traditional resin material, this magnetic resin has increased capacity and is easier to collect from aqueous solutions. $Fe_2O_3$ and $Fe_3O_4$ nanoparticles are, by far, the most widely researched core magnetic supports because of their high conductivity, catalyst-loading capacity, stability, catalytic activity, magnetic susceptibility, and large surface area. Several papers have reported the use of melamine–formaldehyde resin as an effective catalyst in organic reactions. For example, superhydrophilic mesoporous sulfonated melamine–formaldehyde resin-supported palladium was used as a catalyst for biofuel upgrade [10], mesoporous melamine–formaldehyde resin was used for continuous synthesis of cyclic carbonates from epoxides and gaseous $CO_2$ [11], and iron oxide mesoporous structures based on sulfonated melamine–formaldehyde were employed for the a Biginelli reaction [12].

Pyrazoles, five-membered heterocycles, are in the class of bioactive drug targets commonly used in the pharmaceutical sector. They are the building blocks in a majority of biologically active chemicals [13]. For instance, they possess antimicrobial (I) [14], anticancer (II) [15], analgesic (III) [16], and anti-inflammatory (IV) properties (Figure 1) [17]. Various drugs comprising a pyrazolone nucleus have been authorized by the Food and Drug Administration (FDA) to date. In the case of amyotrophic lateral sclerosis (ALS), edaravone V has been employed as a free radical scavenger [18]. In liver function tests, aminophenazone VI, having anti-inflammatory and antipyretic properties, has been employed in breath tests to detect cytochrome P-450 metabolic activity [19]. In people suffering from idiopathic chronic immune thrombocytopenia, eltrombopag VII has been applied to treat low blood platelet levels [20]. Propyphenazone VIII is among the few experimental small compounds involving pyrazolone that have been proposed as therapeutic options [21]. With the aid of numerous catalysts, the production of pyrazolones has been

reported employing a variety of techniques and conditions [13–20]. Although these techniques have certain advantages, they also have drawbacks, such as the use of hazardous bases, a time-consuming workup, catalysts containing transition metals, non-recoverable catalysts, difficult waste disposal, and a longer reaction time. With these considerations in mind, developing a simple, feasible, and ecofriendly synthesis process is of considerable interest.

Thus, in this study, we focused on fabricating magnetic sulfonated melamine–formaldehyde resin (MSMF) as a catalyst for the one-pot, three-component syntheses of benzyl pyrazolyl naphthoquinones and 4-[(indol-3-yl)-arylmethyl]-1-phenyl-3-methyl-5-pyrazolones. These compounds were also tested for antioxidant and antibacterial activities.

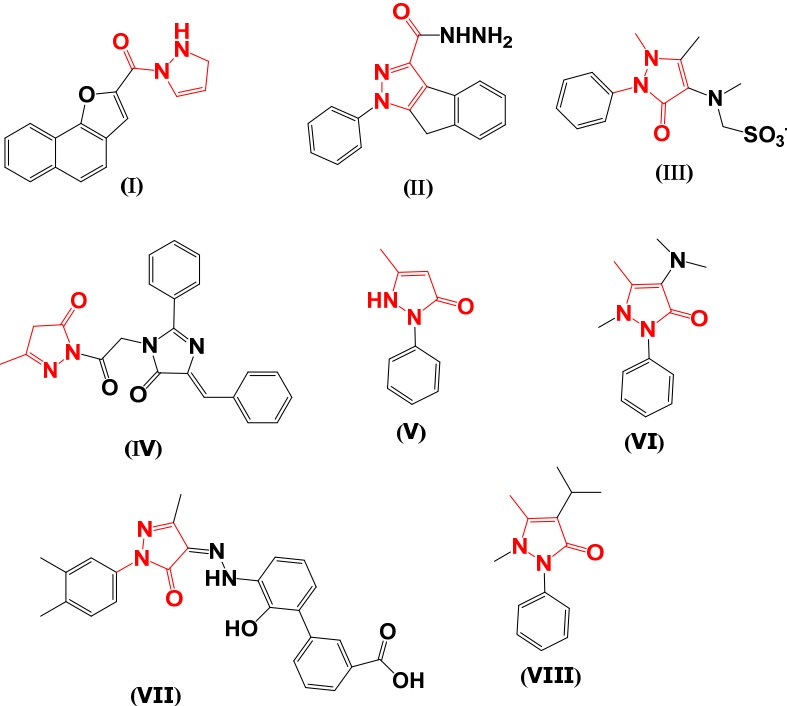

**Figure 1.** Examples of biologically active pyrazolones.

## 2. Results and Discussion

Melamine–formaldehyde resin (MF) offers good physical and chemical stability, as well as numerous secondary amine nitrogen atoms that can be employed as an acidic functionality immobilization site [22]. To form the magnetic sulfonated melamine–formaldehyde, MF was combined with $Fe_3O_4$ magnetite nanoparticles via ultrasonic irradiation at room temperature.

### 2.1. Characterization of Catalysts

**FTIR:** The FTIR absorption spectra of SMF and MSMF are shown in Figure 2A. The SMF and MSMF samples displayed very similar spectra. As shown in Figure 2A, the broad absorption band at 3398 cm$^{-1}$ is related to the stretching vibration of the NH$_2$ and N-H groups. The absorption bands of 1567, 1361, and 1041 cm$^{-1}$ are assigned to the stretching vibrations of the melamine ring, asymmetric O=S=O, and symmetric –SO$_3$H, respectively.

**Elemental analysis:** Elemental analysis was also carried out for the characterization of SMF and MSMF (Table 1). Table 1 shows that the oxygen content of MSMF was higher than that of SMF. The presence of iron in the MSMF sample was indicated by the lower contents of C, H, N, and S. In addition, TGA results (Figure 3) showed that SMF contained

more water (meaning more hydrogen and oxygen) than MSMF; however, a higher content of O in MSMF may indicate the presence of iron oxide.

**Table 1.** Elemental analyses of SMF and MSMF. Values in parentheses indicate expected contents.

| Sample | C% | H% | N% | O% | S% |
|---|---|---|---|---|---|
| SMF | 24.23 (25.36) | 2.98 (3.19) | 28.31 (29.57) | 25.92 (22.52) | 11.09 (11.28) |
| MSMF | 21.08 (21.20) | 2.45 (2.69) | 23.74 (24.91) | 26.30 (22.97) | 9.28 (9.51) |

**X-ray diffraction:** XRD patterns of the bare SMF resin and MSMF are presented in Figure 2B. The XRD patterns of bare SMF and MSMF were nearly identical, with the exception of a slight change between 2 Theta = 10° and 25°, which is due to the presence of $Fe_3O_4$ nanoparticles in the resin. A wide peak was observed in both samples, showing that they both had an amorphous pattern [12].

**VSM:** VSM is a method for examining a material's magnetic properties. The VSM curve of MSMF is shown in Figure 2C. The superparamagnetic property of $Fe_3O_4$ nanoparticles was verified with a magnetization saturation value of 60.57 emu/g. The VSM curve of MSMF, on the other hand, revealed superparamagnetic behavior, with a magnetization saturation (Ms) value of 20 emu/g.

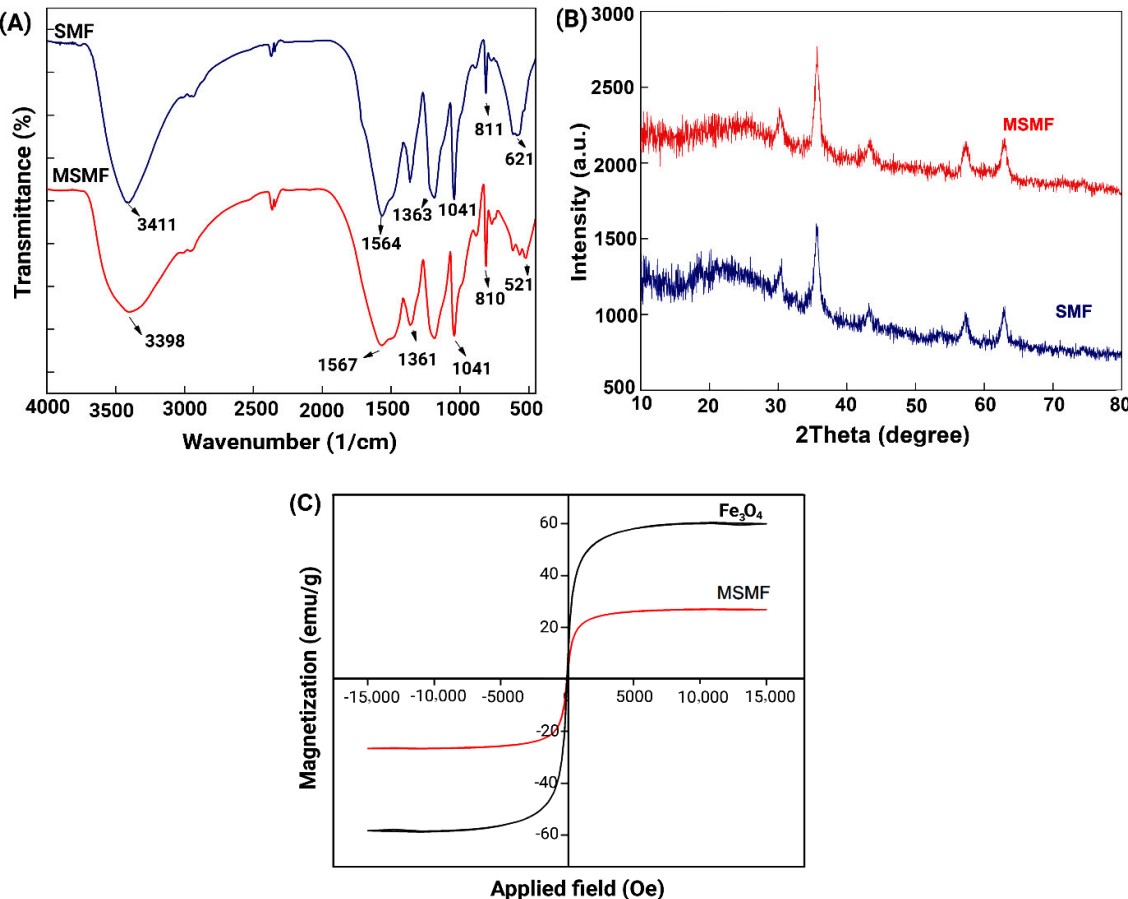

**Figure 2.** FTIR spectra (**A**) and XRD patterns (**B**) of SMF and MSMF. VSM curves of $Fe_3O_4$ and MSMF (**C**).

**TG/DTG**: The TG/DTG curves of SMF and MSMF are presented in Figure 3A,B. Three weight losses were observed in the TG curves of both SMF and MSMF. Evaporation

of adsorbed water on the polymer surface causes weight loss below 100 °C. The decomposition of a portion of melamine causes weight loss in the range of 100~320 °C. The thermal breakdown of the SMF structure is the reason for weight loss in the range of 390–600 °C [23]. Furthermore, due to the presence of Fe₃O₄ nanoparticles, MSMF has higher thermal stability than SMF [24].

**EDX**: The chemical composition of the synthesized SMF and MSMF was determined using the EDX test (Figure 3C). The presence of various quantities of C, O, S, N, and Fe elements was determined by comparing spectra with tabular data. The EDX test confirmed the presence of iron in the MSMF sample.

**FESEM**: Figure 3D shows FESEM images of SMF and MSMF at a magnification of 500 nm. In most areas, the SMF FESEM image revealed an uneven shape and considerable aggregation. On the other hand, the FESEM image of MSMF displayed an aggregation of amorphous particles with diameters of around 40 nm.

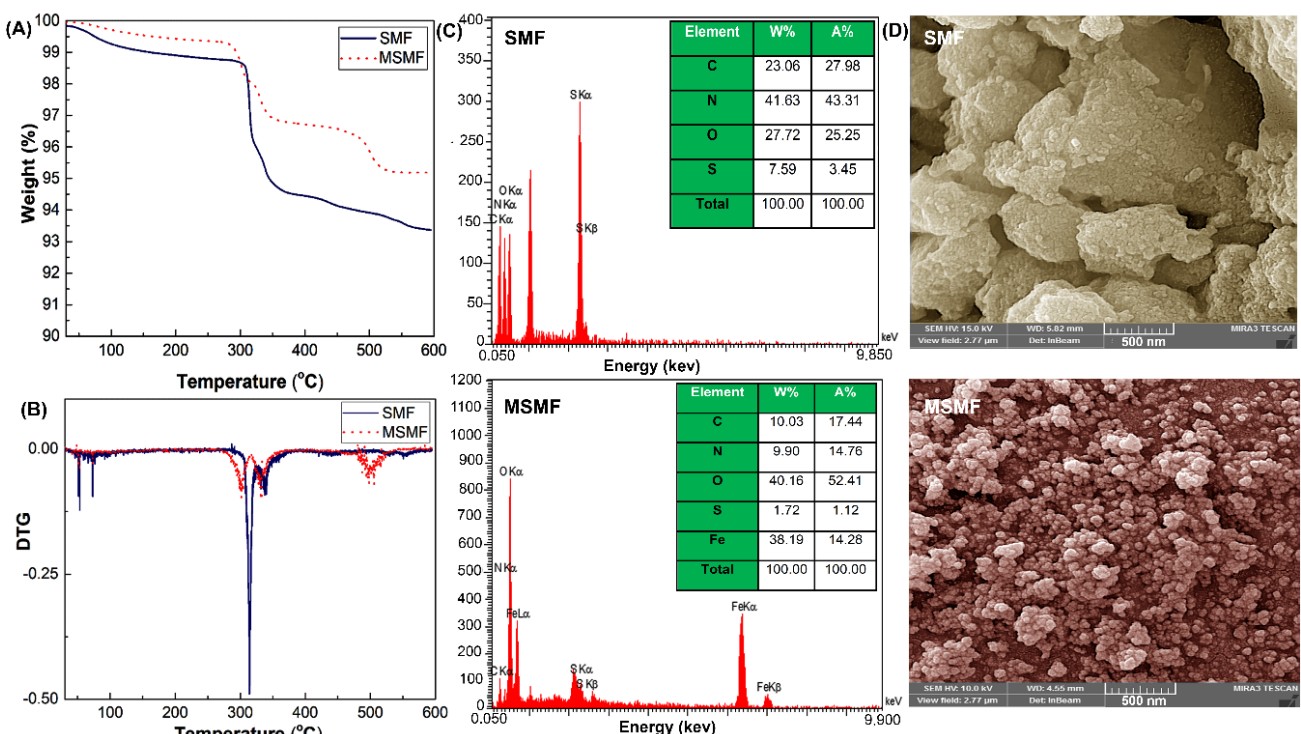

**Figure 3.** TG (**A**) and DTG (**B**) curves, EDX spectra (**C**), and FESEM (**D**) micrographs of SMF and MSMF.

**Ion-exchange capacity (IEC):** Ion-exchange capacity (IEC), which provides an idea of the number of acid groups present in a polymer, can be calculated from the degree of sulfonation (DS) via titration. First, 0.5 g of dry polymer was dissolved in 10 mL of distilled water, which was titrated against standard 0.1 N NaOH solution using phenolphthalein as an indicator. DS and IEC were calculated as [25]:

$$DS = [0.292\ M\ (NaOH) \times V\ (NaOH)] \times 100/[W - 0.08\ M\ (NaOH) \times V\ (NaOH)]$$

$$IEC = 1000 \times DS/[292 + 81 \times DS]$$

where:

M (NaOH) = molarity of standard NaOH solution (mol/L);

V (NaOH) = volume of NaOH solution used to neutralize the polymer solution (mL);

W = sample mass (g);

292= molecular weight of SMF repeat unit; and

81 = molecular weight of the –SO$_3$H

**Acidity test:** The quantity of H$^+$ of the SMF (3.9 mmol/g) and MSMF (3.6 mmol/g) was calculated by acid–base titration. This total acidity includes –SO$_3$H groups based on reaction with NaOH followed by back titration using an HCl solution.

*2.2. Catalytic Activity of MSMF in the Synthesis of 3-Hydroxynaphthalene-1, 4-dione, and 4-[(Indol-3-yl)-arylmethyl]-1-phenyl-3-methyl-5-pyrazolones*

To evaluate the catalytic activity of MSMF, aromatic aldehyde, 3-methyl-1-phenyl-1H-pyrazol-5-ol, and 2-hydroxy naphthoquinone were used to synthesize benzyl pyrazolyl naphthoquinones (Figure 4C).

**Figure 4.** Synthesis of MSMF in two steps (**A**); synthesis of 3-methyl-1-phenyl-1*H*-pyrazol-5-ol (**B**); synthesis of benzyl pyrazolyl naphthoquinones from aldehyde, 3-methyl-1-phenyl-1*H*-pyrazol-5-ol,

and 2-hydroxy naphthoquinone (**C**); synthesis of 4-[(indol-3-yl)-arylmethyl]-1-phenyl-3-methyl-5-pyrazolones from aldehyde, 3-methyl-1-phenyl-1*H*-pyrazol-5-ol, and indole (**D**).

Optimization of the condensation reaction was achieved by the reaction of 3-methyl-1-phenyl-1*H*-pyrazol-5-ol, benzaldehyde, and 2-hydroxy naphthoquinone under various conditions. The maximum yield of the desired benzyl pyrazolyl naphthoquinone was determined with 0.04 g of catalyst at 50 °C in water. Different solvents such as water, CHCl₃, THF, and EtOH affected the time and yield of the reaction. The water solvent achieved the best results for yield and time of the model reaction. The results of the model reaction in different solvents showed that a high-polarity solvent results in a high yield and short reaction time. This reaction was also tested at various temperatures of 25, 50, and 80 °C. The highest yield in the shortest reaction time was observed at 50 °C and with 0.06 g of the catalyst. The results are summarized in Table 2.

**Table 2.** Optimization of the three-component reaction of 3-methyl-1-phenyl-1*H*-pyrazol-5-ol, benzaldehyde, and 2- hydroxy naphthoquinone under various conditions [a].

| Entry | Solvent | Catalyst [c] g | Temp. °C | Time Min. | Yield % [b] |
|---|---|---|---|---|---|
| 1 | EtOH | 0.04 | 50 | 60 | 65 |
| 2 | H₂O | 0.04 | 50 | 40 | 90 |
| 3 | THF | 0.04 | 50 | 180 | 45 |
| 4 | CHCl₃ | 0.04 | 50 | 180 | 60 |
| 5 | Solvent-free | 0.04 | 50 | 120 | 65 |
| 6 | H₂O | 0.04 | r.t | 75 | 50 |
| 7 | H₂O | 0.04 | 80 | 30 | 92 |
| 8 | H₂O | 0.03 | 50 | 60 | 80 |
| 9 | H₂O | 0.06 | 50 | 30 | 94 |
| 10 | H₂O | 0.07 | 50 | 30 | 94 |

[a] Reaction conditions: benzaldehyde (1 mmol), 3-methyl-1-phenyl-1*H*-pyrazol-5-ol (1 mmol), 2-hydroxy naphthoquinone (1 mmol). [b] Isolated yield. [c] Acid catalyst loading in the MSMF is 3.6 mmol/g.

After optimization of the reaction conditions, benzaldehyde derivatives with electron-donating and electron-withdrawing groups were examined in the presence of 0.06 g MSMF (Table 3). The results are shown in Table 3, indicating an excellent yield of benzyl pyrazolyl naphthoquinones obtained under mild conditions in 20–35 min and amounting to 85–95%. In addition, this approach enables the use of 2-thiophencarbaldehyde as a heteroaromatic, as well as other aromatic aldehydes, such as 2-naphtaldehyde, with high yields of benzyl pyrazolyl naphthoquinone in the presence of MSMF.

**Table 3.** Conversion of benzaldehyde derivatives to benzyl pyrazolyl naphthoquinone in the presence of MSMF [a].

| Entry | Product | Product | Time Min. | Yield % [b] | M.P. (°C) Observed | M.P. (°C) Reported | Ref. |
|---|---|---|---|---|---|---|---|
| 1 |  | **4a** | 30 | 94 | 266–267 | 266–268 | [26] |

| 2 |  | **4b** | 25 | 93 | 263–264 | 262–264 | [26] |
|---|---|---|---|---|---|---|---|
| 3 |  | **4c** | 30 | 90 | 264–267 | 264–264 | [26] |
| 4 |  | **4d** | 25 | 92 | 280–281 | 278–280 | [26] |
| 5 |  | **4e** | **20** | 95 | 245–247 | 246–248 | [26] |
| 6 |  | **4f** | 20 | 90 | 253–255 | 252–254 | [26] |
| 7 |  | **4g** | 25 | 90 | 230–232 | 231–233 | [26] |
| 8 |  | **4h** | 25 | 87 | 226–227 | 224–228 | [26] |

| Entry | Structure | Product | Time | Yield | mp | mp (lit.) | Ref. |
|---|---|---|---|---|---|---|---|
| 9 |  | 4i | 20 | 90 | 211–213 | 214–216 | [26] |
| 10 |  | 4j | 35 | 85 | 191–193 | NR | |
| 11 |  | 4k | 20 | 92 | 232–234 | NR | |
| 12 |  | 4l | 15 | 95 | 235–236 | NR | |
| 13 |  | 4m | 20 | 90 | 275–276 | 272–274 | [26] |
| 14 |  | 4n | 35 | 80 | 233–235 | NR | |

[a] Reaction conditions: aldehyde (1 mmol), 3-methyl-1-phenyl-1*H*-pyrazol-5-ol (1 mmol), 2-hydroxy naphthoquinone (1 mmol), water (5 mL), and MSFM (0.06 g) at 50 °C. [b] Isolated yield. NR: not reported.

**Suggested mechanism:** A possible mechanism (Figure 5A) involves the initial activation of aldehyde using the MSMF catalyst, after which 2-hydroxy naphthoquinone attacks the carbonyl group of the activated aldehyde; subsequently, by removing one water molecule, intermediate (**II**) is prepared. The reaction is followed by activation of intermediate (**II**) as a Michael acceptor. Then, Michael's addition of 3-methyl-1-phenyl-1*H*-pyrazol-5-ol to intermediate (**II**) affords (**III**). Intermediate (**III**) converts to (**IV**) via tautomerization, and product (**V**) is obtained after tautomerization. In the second possible mechanism (Figure 5B), the aldehyde is activated using the MSMF catalyst; then, 3-methyl-1-phenyl-1*H*-pyrazol-5-ol attacks the carbonyl group of the activated aldehyde to form intermediate (**VI**). Then, Michael's addition of 2-hydroxy naphthoquinone to intermediate (**VI**) yields the product.

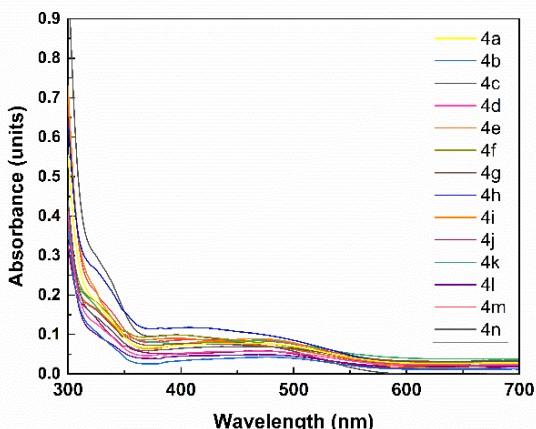

**Figure 5.** Suggested mechanism for the synthesis of benzyl pyrazolyl naphthoquinone catalyzed by MSMF. Ⓟ represents polymer-based catalyst. First possible mechanism (**A**) and second possible mechanism (**B**).

**UV-Visible:** The UV-Vis spectra for eleven benzyl pyrazolyl naphthoquinones in ethanol are shown in Figure 6. Several studies of these molecular systems have been reported, especially 1,4-naphthoquinones, [27]. UV-vis absorption spectra of benzyl pyrazolyl naphthoquinones showed two bands at 201–213 nm and 256–268 nm that are associated with the π–π* electron transitions in the benzene and naphthoquinone, respectively. In addition, a weak band at 325–333 nm was observed that is related to the n–π* transition. Several benzyl pyrazolyl naphthoquinones exhibited wide, low-energy bands in the visible range around 406 and 418 nm [28].

**Figure 6.** UV-vis absorption spectra of benzyl pyrazolyl naphthoquinones in DMSO.

**Comparison of the Catalytic Efficiency of MSMF with Various Catalysts Reported for the Synthesis of Benzyl Pyrazolyl Naphthoquinones:** To show the efficiency of this method in comparison with reported procedures, we selected the reaction of benzalde-

hyde, 2-hydroxy naphthoquinone, and 3-methyl-1-phenyl-1*H*-pyrazol-5-ol for the synthesis of 4a as a model. This comparison is shown in Table 4. It is clear from the data that our method results in shorter reaction times and provides higher yields of the products.

**Table 4.** Comparison of the catalytic efficiency of MSMF with various catalysts reported for the synthesis of 4a.

| Entry | Catalyst | Condition | Yield % | Time Min. | Ref. |
|-------|----------|-----------|---------|-----------|------|
| 1 | MSFM [a] (0.06 g) | $H_2O$/50 °C | 94 | 30 | This study |
| 2 | p-Toluene sulfonic acid (10 mol%) | $H_2O$/Reflux | 80 | 30 | [29] |
| 3 | $MgCl_2$ (20 mol%) | Ethylene glycol/100 °C | 88 | 30 | [30] |
| 4 | - | $H_2O$, MW, 100 °C | 85 | 7 | [31] |
| 5 | β-Cyclodextrin-supported sulfonic acid | $H_2O$, 30 °C | 92 | 40 | [26] |

[a] Acid catalyst loading in the MSMF is 3.6 mmol/g.

Next, various derivatives of 4-[(indol-3-yl)-arylmethyl]-1-phenyl-3-methyl-5-pyrazolones were synthesized with the one-pot condensation of 3-methyl-1-phenyl-1*H*-pyrazol-5-ol, aldehyde, and indole in the presence of a catalytic amount of MSMF (0.05 g) in ethanol at 30 °C (Table 5).

Optimization of the condensation reaction was achieved by the reaction of benzaldehyde, 3-methyl-1-phenyl-1*H*-pyrazol-5-ol, and indole under various conditions. The maximum yield of the desired 4-[(indol-3-yl)-arylmethyl]-1-phenyl-3-methyl-5-pyrazolones (87%) with 0.05 g of catalyst at 30 °C in ethanol was obtained in 80 min. In addition, temperatures of 25, 30, 60, and 80 °C were used to test this reaction. The highest yield was observed in the minimum time at 30 °C and with 0.05 g of catalyst The results are summarized in Table 5.

**Table 5.** Optimization of the three-component reaction of 3-methyl-1-phenyl-1*H*-pyrazol-5-ol, benzaldehyde, and indole [a].

| Entry | Solvent | Catalyst [c] g | Temp. °C | Time Min. | Yield % [b] |
|-------|---------|------------|----------|-----------|-------------|
| 1 | EtOH/$H_2O$ | 0.03 | 30 | 120 | 70 |
| 2 | $H_2O$ | 0.03 | 30 | 180 | 50 |
| 3 | THF | 0.03 | 30 | 180 | 30 |
| 4 | EtOH | 0.03 | 30 | 90 | 75 |
| 5 | $CHCl_3$ | 0.03 | 30 | 180 | 30 |
| 6 | Solvent-free | 0.03 | 30 | 180 | 30 |
| 7 | EtOH | 0.03 | r.t | 120 | 50 |
| 8 | EtOH | 0.03 | 60 | 80 | 40 |
| 9 | EtOH | 0.03 | 80 | 80 | 30 |
| 10 | EtOH | 0.05 | 30 | 80 | 87 |
| 11 | EtOH | 0.07 | 30 | 80 | 88 |

[a] Reaction conditions: benzaldehyde (1 mmol), 3-methyl-1-phenyl-1*H*-pyrazol-5-ol (1 mmol), and indole (1 mmol). [b] Isolated yield. [c] Acid catalyst loading in the MSMF is 3.6 mmol/g.

Following optimization, benzaldehyde derivatives containing electron-donating and electron-withdrawing groups were studied in the presence of 0.05 g MSMF (Table 6).

**Table 6.** Synthesis of 4-[(indol-3-yl)-arylmethyl]-1-phenyl-3-methyl-5-pyrazolones by MSMF and benzaldehyde derivatives [a].

| Entry | Product | Code | Time Min. | Yield % [b] | M.P. (°C) Observed | M.P. (°C) Reported | Ref. |
|-------|---------|------|-----------|-------------|--------------------|--------------------|------|
| 1 | | **6a** | 80 | 87 | 231–233 | 235–236 | [32] |
| 2 | | **6b** | 60 | 90 | 180–183 | 173–175 | [32] |
| 3 | | **6c** | 70 | 87 | 181–183 | 180–182 | [32] |
| 4 | | **6d** | 65 | 88 | 151–153 | 170–171 | [32] |
| 5 | | **6e** | 50 | 93 | 183–185 | 184–186 | [32] |

| 6 |  | **6f** | 60 | 90 | 238–239 | 242–244 | [32] |
| 7 |  | **6g** | 65 | 89 | 194–196 | 206 | [33] |
| 8 |  | **6h** | 70 | 89 | 164–166 | 161–163 | [32] |
| 9 |  | **6i** | 60 | 91 | 195–197 | 191–193 | [32] |
| 10 |  | **6j** | 50 | 90 | 242–244 | NR | |
| 11 |  | **6k** | 90 | 85 | 244–246 | 246 | [33] |

| | | | | | | |
|---|---|---|---|---|---|---|
| 12 |  | **6l** | 70 | 87 | 210–212 | NR |
| 13 |  | **6m** | 50 | 83 | 193–195 | NR |
| 14 |  | **6n** | 35 | 90 | 231–233 | NR |
| 15 |  | **6o** | 40 | 91 | 185–187 | NR |
| 16 |  | **6p** | 50 | 85 | 185–187 | NR |

[a] Reaction conditions: aldehyde (1 mmol), 3-methyl-1-phenyl-1*H*-pyrazol-5-ol (1 mmol), indole (1 mmol), (3 mL) EtOH and MSFM (0.05 g) at 30 °C. [b] Isolated yield. NR: not reported.

**Suggested mechanism:** In the suggested mechanism (Figure 7), first, the aldehyde is activated using the MSMF catalyst. The indole then attacks the activated aldehyde's carbonyl group, forming intermediate (**II**) by removing one water molecule. Michael's reaction of 3-methyl-1-phenyl-1*H*-pyrazol-5-ol with intermediate (**II**) produces intermediate (**III**). The tautomerization process converts intermediate (**III**) to product (**IV**).

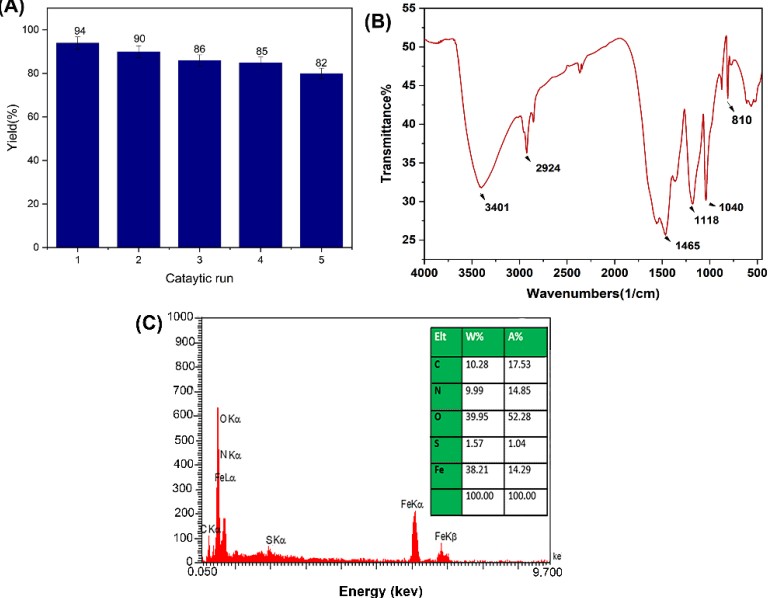

**Figure 7.** Suggested mechanism for the synthesis of 4-[(indol-3-yl)-arylmethyl]-1-phenyl-3-methyl-5-pyrazolones catalyzed by MSMF. Ⓟ represents polymer-based catalyst.

**Recovery:** The reuse and recycling of the MSMF catalyst for the synthesis of benzyl pyrazolyl naphthoquinones were also studied (Figure 8). The catalyst was recovered by adding water and ethanol to the reaction mixture in the synthesis of 4a as a representative model after running the reaction under optimized conditions, and it was isolated by an external magnet. The catalyst was rinsed in ethanol (5 mL) before being dried under decreased pressure and was then added back to the process. Even after five times, the MSMF catalyst showed high catalytic activity. The recycled catalyst was identified by FT-IR and EDX analyses. Comparison of the FT-IR spectrum of MSMF before and after five consecutive runs confirmed that no definite change in its structure was observed; therefore, MSMF can be considered a recyclable and stable catalyst in organic reactions (Figure 8B). However, EDX analysis of the recovered catalyst (Figure 8C) showed that a degree of catalyst desulfation and leaching occurred after five runs, which explains the decrease in the yield of reactions.

**Figure 8.** Reuse of MSMF in the synthesis of benzyl pyrazolyl naphthoquinones (**A**), as well as FTIR (**B**) and EDX (**C**) spectra of the recovered MSMF catalyst.

**Antioxidant activity**: The antioxidant activity of Fe₃O₄ nanoparticles, SMF, MSMF, benzyl pyrazolyl naphthoquinones, and 4-[(indol-3-yl)-arylmethyl]-1-phenyl-3-methyl-5-pyrazolones was studied using the 2,2-diphenyl-1-picrylhydrazyl (DPPH) scavenging model (Figure 9). Fe₃O₄ nanoparticles, SMF, and MSMF showed antioxidant activity levels of 87%, 65%, and 78%, respectively, due to the presence of hydroxyl and sulfonyl groups in their structures. In addition, the presence of hydroxyl and amine groups in the structure of synthesized derivatives led to antioxidant activity of between 40% and 90%.

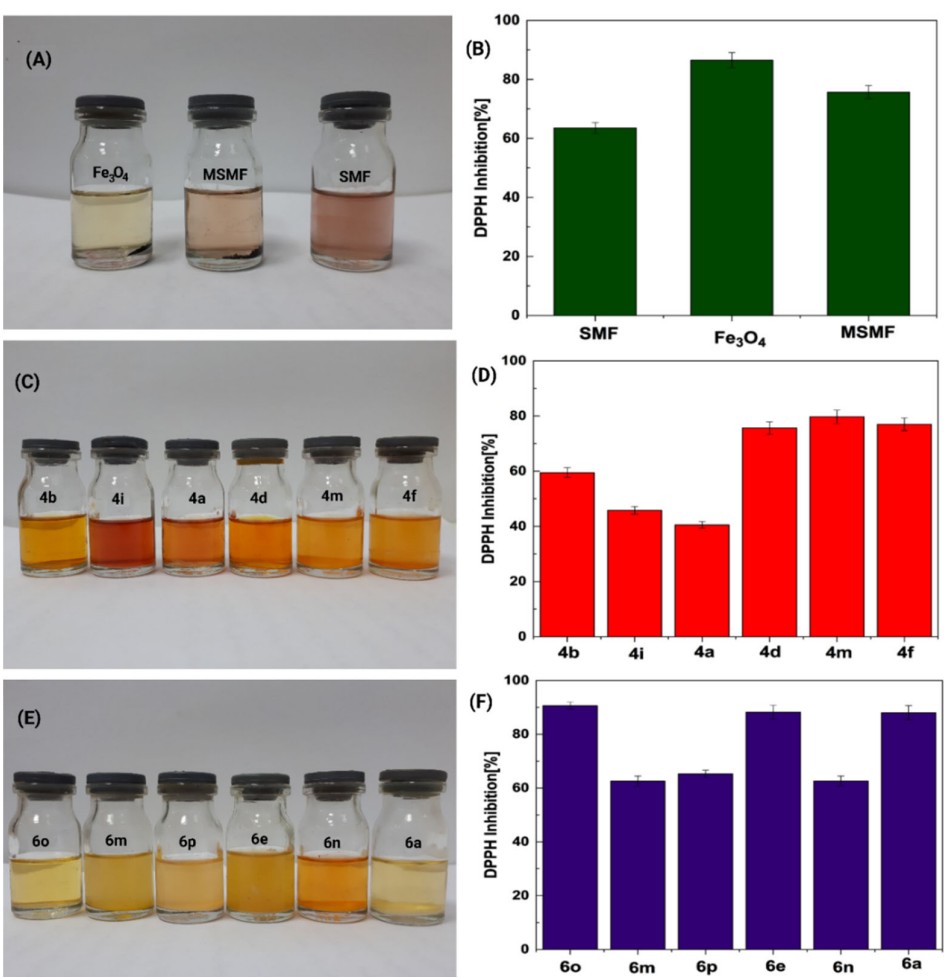

**Figure 9.** Photographs (**A**,**C**,**E**), and histograms of antioxidant activities of Fe₃O₄ nanoparticles, SMF, MSMF (**B**), synthesized benzyl pyrazolyl naphthoquinone (**D**), and 4-[(indol-3-yl)-arylmethyl]-1-phenyl-3-methyl-5-pyrazolones (**F**).

**Antibacterial activity:** The in vitro antibacterial activities of the Fe₃O₄ nanoparticles, SMF, MSMF, benzyl pyrazolyl naphthoquinone (4a, 4b, 4d, and 4i), and 4-[(indol-3-yl)-arylmethyl]-1-phenyl-3-methyl-5-pyrazolones (6e, 6m, 6p, and 6o) were investigated against *Escherichia coli* and *Staphylococcus aureus*; results are shown in Figure 10 and Table 7. The MSMF sample had the best antibacterial efficacy against both microorganisms, whereas the SMF sample was only efficient against *Escherichia coli*. Moreover, the benzyl pyrazolyl naphthoquinones and 4-[(indol-3-yl)-arylmethyl]-1-phenyl-3-methyl-5-pyra-zolones showed antibacterial growth-inhibitory effects against *Staphylococcus aureus*.

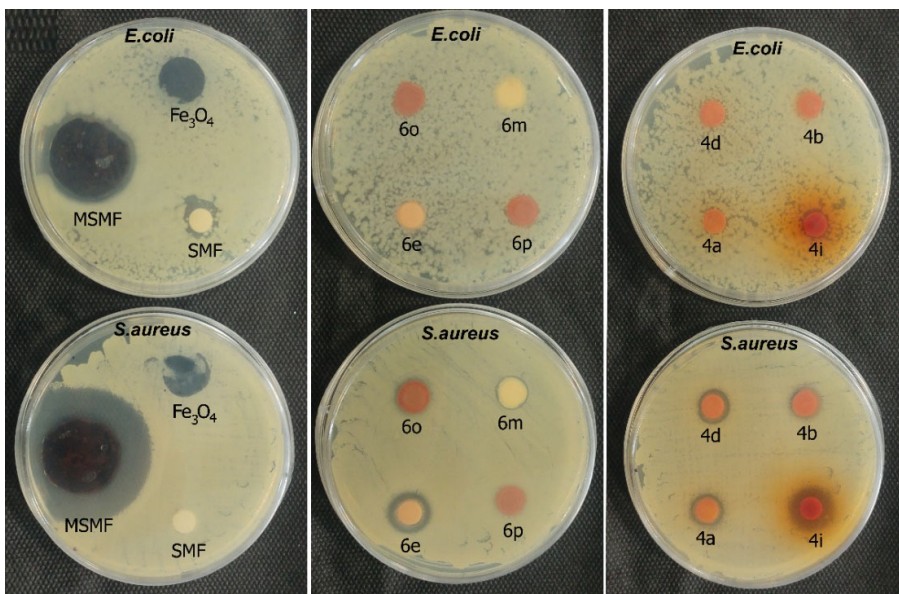

**Figure 10.** Antibacterial activities of SMF, MSMF, some benzyl pyrazolyl naphthoquinone, and 4-[(indol-3-yl)-arylmethyl]-1-phenyl-3-methyl-5-pyrazolones against *Escherichia coli* and *Staphylococcus aureus* via Kirby–Bauer disk diffusion technique.

**Table 7.** Antibacterial activities of catalysts and some benzyl pyrazolyl naphthoquinone, and 4-[(indol-3-yl)-arylmethyl]-1-phenyl-3-methyl-5-pyrazolones via Kirby–Bauer disk diffusion technique.

| | | Inhibition Zone (mm) | |
|---|---|---|---|
| **Entry** | **Compound** | *Staphylococcus aureus* Gram-Positive (+) | *Escherichia coli* Gram-Negative (−) |
| 1 | Fe₃O₄ NPs | 16 ± 1.2 | NE [a] |
| 2 | SMF | NE | 12 ± 1.7 |
| 3 | MSMF | 40 ± 1.2 | 27 ± 1.7 |
| 4 | 4a | 11 ± 0.6 | 8 ± 07 |
| 5 | 4b | 11 ± 0.7 | 10 ± 1.2 |
| 6 | 4d | 11 ± 1.0 | NE |
| 7 | 4i | 15 ± 1.0 | 9 ± 07 |
| 8 | 6e | 10 ± 1.0 | NE |
| 9 | 6o | NE | NE |
| 10 | 6m | 14 ± 0.7 | NE |
| 11 | 6p | NE | NE |
| 12 | Gentamicin | 26 ± 1.2 | 19.6 ± 0.7 |
| 13 | Chloramphenicol | 22.3 ± 1.7 | 20.7 ± 1.0 |

[a] No effect.

## 3. Experimental Procedures

### 3.1. Materials

Melamine, formaldehyde, ferrous chloride (FeCl₂·4H₂O), ferric chloride (FeCl₃·6H₂O), sodium hydrogen sulfite, sodium hydroxide, and all solvents were purchased from Sigma-Aldrich. Indole, 2-hydroxy naphthoquinone, phenylhydrazine, ethyl acetoacetate, and all other reagents were provided by Merck Company, Germany.

### 3.2. Preparation of Fe₃O₄ Magnetic Nanoparticles

FeCl₃·6H₂O (2.1 mmol) and FeCl₂·4H₂O (6.3 mmol) with a molar ratio of 1:3 were dissolved in deionized water. The mixture was stirred mechanically at 80 °C for 15 min, and

$NH_4OH$ (10 mL, 25%) was quickly added until the pH reached 10. After 30 min, the black precipitate was separated via magnetic decantation and washed several times with deionized water and twice with ethanol. The $Fe_3O_4$ nanoparticles were characterized using FTIR, SEM/EDX, and XRD analyses (Figure S1 in Supplementary Materials).

### 3.3. Synthesis of Magnetic Sulfonated Melamine–Formaldehyde Resin

Magnetic sulfonated melamine–formaldehyde resin (MSMF) was fabricated in two steps, as follows:

A three-neck, round-bottomed flask supplied with a mechanical stirrer was filled with melamine (50 g, 0.39 mol), formaldehyde (36 g, 1.2 mol), and distilled water (200 mL). The pH of the solution was adjusted to 8.5–9.0 using sodium hydroxide (1.5 g, 0.03 mol) after the melamine was dissolved, and the solution was gently heated to 80 °C. $NaHSO_3$ (3.1 g, 0.03 mol) was added to the solution, and the pH was adjusted to 4–5 [10]. After 40–50 min, the pH of the solution was adjusted to 3–4 using sulfuric acid solution (3 N), and the solution was heated to 80 °C for 5 h; sulfonated melamine–formaldehyde (SMF) was obtained as a colorless powder after the mixture was filtered, washed with petroleum ether and water, and then dried at room temperature. The MSMF material was made in the second stage by refluxing the mixture of SMF (0.4 g) and $Fe_3O_4$ nanoparticles (0.057 g) in distilled water for 4 h. After stirring for 12 h at room temperature, the product was separated with a magnet, rinsed with deionized water and ethanol, and dried overnight at 60 °C (Figure 4A).

### 3.4. General Procedure for the Synthesis of 3-Methyl-1-phenyl-1H-pyrazol-5-ol

The suggested one-pot approach or the traditional two-step procedure can be used to synthesize 4-arylidenepyrazolones. The synthesis of 3-methyl-1-phenyl-5-pyrazolone was performed by combining ethyl acetoacetate (0.05 mol, 6.2 mL) with phenylhydrazine (0.05 mol, 5 mL) in a standard process. Then, 0.5 mL of acetic acid was added, followed by stirring for 1 h at 90 °C. After allowing the heavy syrup to cool, ether (10–20 mL) was added to the mixture and rapidly stirred to obtain crystalline pyrazolone. The isolated product was filtered and further purified using ethanol recrystallization (Figure 4B).

### 3.5. Synthesis of 3-Hydroxynaphthalene-1, 4-dione (Benzylpyrazolyl Naphthoquinone) Derivatives

A mixture of 3-methyl-1-phenyl-1*H*-pyrazol-5-ol (1 mmol), an aldehyde (1.0 mmol), 2-hydroxy naphthoquinone (1 mmol), and MSMF (0.06 g) in water (5 mL) was stirred at 50 °C and monitored via thin-layer chromatography (hexane/ethyl acetate 5:1) until the reaction was complete. Afterward, the product was separated by adding 3 mL of ethanol to the mixture and then filtering it. The catalyst was separated with an external magnet. Finally, the separated product was washed twice with ethanol (2 × 5 mL) (Figure 4C).

### 3.6. Synthesis of 4-[(Indol-3-yl)-arylmethyl]-1-phenyl-3-methyl-5-pyrazolones

A mixture of 3-methyl-1-phenyl-1*H*-pyrazol-5-ol (1.0 mmol), an aldehyde (1.0 mmol), indole (1.0 mmol), and MSMF (0.05 g) in ethanol (3 mL) was stirred at 30 °C and monitored via thin-layer chromatography (hexane/ethyl acetate 4:1) until the reaction was complete. The catalyst was separated with an external magnet. The crude solid product was filtered and washed twice with ethanol (2 × 5 mL) (Figure 4D).

### 3.7. Characterization

Fourier transform infrared spectroscopy (FTIR, Bruker Tensor 27, Bremen, Germany), hydrogen and carbon nuclear magnetic resonance spectroscopy (1HNMR and 13CNMR, Bruker Avance DRX-400, Bremen, Germany), and a CHNSO elemental analyzer (ECS8020 Costech, Milan, Italy) were employed for the chemical characterization of the products. The crystallinity and surface morphology of samples were investigated via X-

ray diffraction (XRD, Shibuya-ku, Tokyo, Japan) and field-emission scanning electron microscopy (FESEM, Hitachi S 4160, Hitachi, Japan). Ultraviolet–visible spectroscopy (UV-Vis, Cecil 5000 series, Cambridge, UK) was applied for evaluation of the antioxidant and optical activity of samples. Thermogravimetric analysis (TGA 209F3, NETZSCH, Selb, Germany) was applied for the investigation of the thermal stability of samples. Mass spectrometry (5975c-inert MSD with Triple-Axis Detector, Agilent, Santa Clara, FL, USA) was used to quantify products.

## 4. Conclusions

A magnetic sulfonated melamine–formaldehyde resin material was fabricated from a mixture of sulfonated melamine–formaldehyde and $Fe_3O_4$ nanoparticles in two steps and was used as a catalyst in the synthesis of benzyl pyrazolyl naphthoquinones, as well as 4-[(indol-3-yl)-arylmethyl]-1-phenyl-3-methyl-5-pyrazolones, via one-pot, three-component condensation. FESEM and XRD analyses showed an amorphous structure, with an average particle size of 40 nm for magnetic catalysts. Compared with sulfonated melamine–formaldehyde resin, the magnetic catalyst had higher thermal stability. The maximum yield of the benzyl pyrazolyl naphthoquinone with 0.04 g of catalyst at 50 °C in water as a green solvent and mild conditions was obtained at 95% in 40 min. On the other hand, the maximum yield of the 4-[(indol-3-yl)-arylmethyl]-1-phenyl-3-methyl-5-pyrazolones with 0.05 g of catalyst at 30 °C in ethanol was obtained at 87% in 80 min. The magnetic catalyst was able to be reused at least five times without a substantial loss of catalytic activity. Interestingly, the presence of hydroxyl and amine groups in the produced benzyl pyrazolyl naphthoquinones and 4-[(indol-3-yl)-arylmethyl]-1-phenyl-3-methyl-5-pyrazolones resulted in antioxidant activity ranging from 40% to 90%. Moreover, the benzyl pyrazolyl naphthoquinones and 4-[(indol-3-yl)-arylmethyl]-1-phenyl-3-methyl-5-pyrazolones showed antibacterial growth-inhibitory effects against *Staphylococcus aureus*.

**Supplementary Materials:** The following supporting information can be downloaded at: www.mdpi.com/article/10.3390/catal12060626/s1, Figure S1: FTIR spectrum (A), EDX spectrum (B), XRD pattern (C), and SEM image (D) of $Fe_3O_4$ nanoparticles, Figure S2: mixture of the sulfonated melamine–formaldehyde (SMF) in distilled water and mixture of SMF and $Fe_3O_4$ nanoparticles (A); magnetic properties of $Fe_3O_4$ nanoparticles and MSMF (B), Figure S3: $^1$H-NMR spectra of 4e, Figure S4: $^{13}$C-NMR spectra of 4e, Figure S5: Mass spectra of 4e, Figure S6: $^1$H-NMR spectra of 4j, Figure S7: 13C-NMR spectra of 4j, Figure S8: Mass spectra of 4j, Figure S9: $^1$H-NMR spectra of 4k, Figure S10: $^{13}$C-NMR spectra of 4k, Figure S11: Mass spectra of 4k, Figure S12: $^1$H-NMR spectra of 4l, Figure S13: $^1$H-NMR spectra of 4l, Figure S14: Mass spectra of 4l, Figure S15: $^1$H-NMR spectra of 4n, Figure S16: $^{13}$C-NMR spectra of 4n, Figure S17: Mass spectra of 4n, Figure S18: $^1$H-NMR spectra of 6j, Figure S19: $^{13}$C-NMR spectra of 4j, Figure S20: Mass spectra of 4o, Figure S21: $^1$H-NMR spectra of 6l, Figure S22: $^{13}$C-NMR spectra of 6l, Figure S23: Mass spectra of 6l, Figure S24: $^1$H-NMR spectra of 6m, Figure S25: $^{13}$C-NMR spectra of 6m, Figure S26: Mass spectra of 6m, Figure S27: $^1$H-NMR spectra of 6n, Figure S28: $^{13}$C-NMR spectra of 6n, Figure S29: Mass spectra of 6n, Figure S30: $^1$H-NMR spectra 6o, Figure S31: $^{13}$C-NMR spectra of 6o, Figure S32: Mass spectra of 6O, Figure S33: $^1$H-NMR spectra 6p, Figure S34: $^{13}$C-NMR spectra of 6p, Figure S35: Mass spectra of 6p.

**Author Contributions:** Methodology, S.M.N.; investigation, S.M.N.; writing—original draft preparation, E.N.Z.; writing—review and editing, E.N.Z., S.A.P., G.H., and P.M.; supervision, S.A.P. and E.N.Z.; project administration, E.N.Z. and S.A.P. All authors have read and agreed to the published version of the manuscript.

**Funding:** This research received no external funding.

**Data Availability Statement:** The datasets used and/or analyzed during the current study are available from the corresponding author on reasonable request.

**Acknowledgments:** E.N.Z. and S.A.P. are thankful to Damghan University for financial support of the current research.

**Conflicts of Interest:** The authors declare no conflict of interest.

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
