# Peer review of "Magnetic Sulfonated Melamine-Formaldehyde Resin as an Efficient Catalyst for the Synthesis of Antioxidant and Antimicrobial Pyrazolone Derivatives"

_catalysts, doi:10.3390/catal12060626_

Round 1
Reviewer 1 Report
The authors present interesting new results on the synthesis and characterization of new catalyst. The manuscript is well written. I recommend to accept the manuscript with minor revision.
I recommend to estimate, on the grounds of CHNOS analysis (page 6), the composition of SMF and MSMF - the ratio of monomers, content of Fe3O4 in MSMF, content of water. Else these figures (elements content values) do not provide significant information for the reader. Will these estimations agree with the results of ion-exchange capacity measurements (page 7)?
I also have just one technical comment: in Supporting materials, captions for Figures S4, S5, S7, S8, S10 and some others - the authors write "spectra", however it should be a single "spectrum".
Author Response
The authors present interesting new results on the synthesis and characterization of new catalyst. The manuscript is well written. I recommend to accept the manuscript with minor revision.
- I recommend to estimate, on the grounds of CHNOS analysis (page 6), the composition of SMF and MSMF - the ratio of monomers, content of Fe3O4 in MSMF, content of water. Else these figures (elements content values) do not provide significant information for the reader. Will these estimations agree with the results of ion-exchange capacity measurements (page 7)?
Response: Thank you so much for your comment. In this work, we used the CHNS, TGA/DTG, VSM and EDX for confirming the presence of iron oxide in the catalyst. Indeed, we need an XPS analysis for final confirmation. Unfortunately, there is one XPS device in Iran. To do so, we have to ship the samples to Tehran. It takes between three and six months to get the results of the analysis, which is a very long time. It is highly appreciated if you consider this. On the other hand, it is very difficult to estimate the number of reactive monomers in the final product of the polymer because the polymer is a large molecule with a high molecular weight. In addition, the best technique for characterizing a polymer is NMR, due to the insolubility of the polymer in solvent we could not carry out this analysis. On the other hand, in CHNSO analysis, we can estimate the theoretical values and compare theoretical values with found data (according to CHNS) (Please see Table 1 in the revised manuscript). In addition, the Ion-exchange capacity (IEC) on page 7 is an approximate method and its data are not comparable with elemental analysis data.
Table 1. Elemental analyses of SMF and MSMF. Found (Expected).
|
S% |
O% |
N% |
H% |
C% |
Samples |
|
11.09(11.28) |
25.92(22.52) |
28.31(29.57) |
2.98(3.19) |
24.23(25.36) |
SMF |
|
9.28(9.51) |
26.30(22.97) |
23.74(24.91) |
2.45(2.69) |
21.08(21.20) |
MSMF |
- I also have just one technical comment: in Supporting materials, captions for Figures S4, S5, S7, S8, S10, and some others - the authors write "spectra", however it should be a single "spectrum".
Response: Thank you for this comment. It was corrected. Please see the revised Supporting materials.
Reviewer 2 Report
This contribution describes the synthesis of magnetic sulfonated melamine–formaldehyde resin (MSMF) and its application as a heterogeneous catalyst for the one-pot, three-component condensation of benzyl pyrazolyl naphthoquinones and 4-[(indol-3-yl)-arylmethyl]-1-phenyl-3-methyl-5-pyrazolones. Magnetic sulfonated melamine–formaldehyde resin (MSMF) was fabricated in two steps: 1. the condensation between melamine and formaldehyde; 2. Incorporation of Fe3O4 nanoparticles. Detailed characterizations of MSMF catalysts were performed via FTIR spectroscopy, elemental analysis, X-ray diffraction, thermogravimetric analysis, field-emission scanning electron microscopy (FESEM), etc. Through acid-base back titration, the number of H+ of the SMF (3.9 mmol/g) and MSMF (3.6 mmol/g) was confirmed. A series of benzyl pyrazolyl naphthoquinones and 4-[(indol-3-yl)-arylmethyl]-1-phenyl-3-methyl-5-pyrazolones derivatives were synthesized using MSMF catalysts in a one-pot, three-component condensation. In the final section, the antioxidant and antibacterial activity of MSMF, synthesized benzyl pyrazolyl naphthoquinones, and 4-[(indol-3-yl)-arylmethyl]-1-phenyl-3-methyl-5-pyrazolones derivatives were tested. Most of the experiments are well performed, even though a few important questions/comments needed to be addressed (listed bellowed). Overall, I believe this work would be of general interest to the readership of Catalyst. I would recommend to accepting this work with revision.
1. Melamine–formaldehyde resin is a widely used thermoset polymer, however, is not famous for catalytic application. In the third paragraph of introduction, the previous reports about the catalytic application of Melamine–formaldehyde resin are missing, such as (ACS Sustainable Chem. Eng. 2020, 8, 12852−12869). I would encourage authors to enhance it.
2. I would like to know if a one-pot condensation, four-component synthesis from phenyl hydrazine, ethyl acetoacetate (including reaction from Fig. 2-B into one-pot reaction) is possible using MSMF catalyst.
3. What is the polydispersity of the MSMF catalyst? Can the author measure the uniformity of the catalyst, such as using dynamic light scattering?
4. I encourage the author to indicate the catalytic entity loading, the acid, in all of the catalytic experiments. For example, in Table 2, table 4 and table 5, the authors only demonstrated the amount (weight in gram) of the MSMF, however, the acid loading is essential to compared with previous reports.
5. Additionally, I would encourage the authors to better interpret their catalytic results from control experiments rather than listing results in the table. For example, in table 2, compare entry 1, 2, 3 and 4, why water is best solvent for this transformation?
6. I encourage the authors to provide a kinetic plot/time course monitor the formation of intermediate and product and the consumption of starting materials, which may be helpful to figure out which is the more possible mechanism in Figure 5.
7. Can the author explain why water is the best solvent for the synthesis of benzyl pyrazolyl naphthoquinones while EtOH is the best solvent for synthesis of 4-[(indol-3-yl)-arylmethyl]-1-phenyl-3-methyl-5-pyrazolones?
8. In the section of Recovery on page 16, characterizations of recovered MSMF catalyst should be provided and further compared with the unused catalyst. Particularly, there is sightly drop on yields on Run #5 than Run #1 in Figure 8. So, a detailed analysis of the reaction residual after catalyst isolation would be helpful.
9. Please fix the misspell of ‘Yield’ on the Y-axis in Figure 8.
10. I think testing the antioxidant and antibacterial activity of the synthesized benzyl pyrazolyl naphthoquinones, and 4-[(indol-3-yl)-arylmethyl]-1-phenyl-3-methyl-5-pyrazolones derivatives is usefully. However, I am confused why to test the antioxidant and antibacterial activity of the MSMF catalyst. Could the authors explain more about this?
11. The authors provide comprehensive characterizations of the synthesized compounds, however the 1H-NMR and 13C-NMR spectra are in relatively low resolution. Please fix the spectrums in supplementary materials if possible.
12. One advantage of this work is to use water as solvent. I encourage author to emphasize in the abstract and summary.

Author Response
The file of response to reviewer is attached.

Round 2
Reviewer 2 Report
Thank you for the response to my comments. I am pleased with most of the author responses to specific questions, they have clarified most of those. However, for comment #4, I still encourage the authors to indicate the acid catalyst loading in the table for audiences' convenience.
Author Response
Thank you for your comment the acid catalyst loading was specified in the Tables 2, 4 and 5 and highlighted in the revised manuscript.